# Pre-Synaptic GABA_A_ in NaV1.8^+^ Primary Afferents Is Required for the Development of Punctate but Not Dynamic Mechanical Allodynia following CFA Inflammation

**DOI:** 10.3390/cells11152390

**Published:** 2022-08-03

**Authors:** Sheng Liu, Veronica Bonalume, Qi Gao, Jeremy Tsung-Chieh Chen, Karl Rohr, Jing Hu, Richard Carr

**Affiliations:** 1Department of Pathology, Affiliated Hospital of Zunyi Medical University, Zunyi 563003, China; sheng.liu@pharma.uni-heidelberg.de; 2Department of Pharmacology, Heidelberg University, 69120 Heidelberg, Germany; 3Department of Pharmacological and Biomolecular Sciences, Università degli Studi di Milano, 20133 Milan, Italy; veronica.bonalume@unimi.it; 4Core Facility Platform Mannheim, Medical Faculty Mannheim, Heidelberg University, 68167 Mannheim, Germany; qi.gao@medma.uni-heidelberg.de; 5Biomedical Computer Vision Group, BioQuant, IPMB, Heidelberg University, 69120 Heidelberg, Germany; k.rohr@uni-heidelberg.de; 6Center for Interdisciplinary Pain Medicine, Department of Anesthesiology, University Hospital of Würzburg, 97080 Würzburg, Germany; chen_j1@ukw.de; 7Experimental Pain Research, Mannheim Centre for Translational Neuroscience (MCTN), Medical Faculty Mannheim, Heidelberg University, 68167 Mannheim, Germany

**Keywords:** GABA_A_, presynaptic inhibition, NaV1.8, spinal dorsal horn, punctate mechanical allodynia, c-fos, electrical excitability, activity-dependent slowing

## Abstract

Hypersensitivity to mechanical stimuli is a cardinal symptom of neuropathic and inflammatory pain. A reduction in spinal inhibition is generally considered a causal factor in the development of mechanical hypersensitivity after injury. However, the extent to which presynaptic inhibition contributes to altered spinal inhibition is less well established. Here, we used conditional deletion of GABA_A_ in NaV1.8-positive sensory neurons (*Scn10a^Cre^*;*Gabrb3^fl/fl^*) to manipulate selectively presynaptic GABAergic inhibition. Behavioral testing showed that the development of inflammatory punctate allodynia was mitigated in mice lacking pre-synaptic GABA_A_. Dorsal horn cellular circuits were visualized in single slices using stimulus-tractable dual-labelling of *c-fos* mRNA for punctate and the cognate c-Fos protein for dynamic mechanical stimulation. This revealed a substantial reduction in the number of cells activated by punctate stimulation in mice lacking presynaptic GABA_A_ and an approximate 50% overlap of the punctate with the dynamic circuit, the relative percentage of which did not change following inflammation. The reduction in dorsal horn cells activated by punctate stimuli was equally prevalent in parvalbumin- and calretinin-positive cells and across all laminae I–V, indicating a generalized reduction in spinal input. In peripheral DRG neurons, inflammation following complete Freund’s adjuvant (CFA) led to an increase in axonal excitability responses to GABA, suggesting that presynaptic GABA effects in NaV1.8^+^ afferents switch from inhibition to excitation after CFA. In the days after inflammation, presynaptic GABA_A_ in NaV1.8^+^ nociceptors constitutes an “open gate” pathway allowing mechanoreceptors responding to punctate mechanical stimulation access to nociceptive dorsal horn circuits.

## 1. Introduction

Patients with chronic neuropathic and inflammatory pain commonly report a heightened sensitivity to mechanical stimuli that disrupts daily life [1]. In people, tactile allodynia is mediated by large diameter Aβ mechanoreceptors that gain access to central pain pathways after injury [2]. Rodent models of neuropathic pain also indicate that inflammation disinhibits polysynaptic pathways in the spinal cord. This allows low threshold mechanoreceptor (LTMR) Aβ and Aδ afferents terminating in dorsal horn laminae III–V to engage projection neurons in superficial lamina I [3] and lamina II [4,5,6]. These poly-synaptic circuits with ventral to dorsal orientation comprise different sets of segmental interneurons for different forms of low threshold mechanical input and are subject to both post-synaptic and presynaptic modulation [7].

Presynaptic inhibition is a potent form of inhibition that occurs at the first order synapse of primary afferent A and C-fibre terminals upstream of the dorsal horn circuitry. Canonical presynaptic inhibition is mediated primarily by GABA release at axo-axonic synapses [8,9]; however, glutamate [10,11], dopamine [12], and opioids can also exert presynaptic effects [13]. A structural basis for presynaptic inhibition is provided by glomeruli in the outer dorsal horn laminae. Glomeruli comprise a central afferent terminal encircled by axo-axonic and dendro-axonic contacts [14]. A-fibres terminate in glomeruli surrounded by GABAergic and glycinergic axo-axonic contacts [15] comprised of inhibitory interneurons, for example, those expressing the calcium binding protein parvalbumin (PV) and parvalbumin-expressing cells that form axoaxonic synapses with A-fibre mechanoreceptor afferent terminals [16]. Nonpeptidergic C-fibre afferents terminate in glomeruli in lamina IIi and receive predominantly GABAergic synaptic contacts [15]. Peptidergic C-fibres terminate in lamina Iio largely devoid of axo-axonic contacts [17] but may nevertheless be subject to modulation by GABA through volume transmission [18].

GABAergic presynaptic inhibition is mediated by ionotropic GABA_A_ receptors that are permeable to chloride and bicarbonate and expressed in the central terminals of A and C fibres, their cell bodies [19,20], and along their peripheral axons [21]. In contrast to CNS neurons, GABA_A_ activation in DRG neurons produces an inward current and thus depolarization [22], which, in primary afferent terminals, leads to inhibition through two primary processes. Firstly, depolarisation reduces the number of available voltage-gated sodium channels (NaV) in the terminal. Secondly, GABA_A_ opening increases membrane conductance. Both processes reduce the amplitude and rate of change of membrane potential during action potential invasion of the terminal and thus, quell transmitter release.

The chloride reversal potential (E_Cl_) in DRG neurons varies widely, ranging from −20 mV to −70 mV [23,24,25], and E_Cl_ contributes to the depolarized reversal potential for GABA in DRG neurons. The intracellular chloride concentration in mature DRG neurons is set by the inward chloride transporter NKCC1, with little to no expression of outward chloride transporters, including KCC2 [26]. Importantly, inflammation leads to the phosphorylation of NKCC1 and an elevated E_Cl_ [27] and thus, larger depolarising responses to GABA in DRG neurons [28].

As postulated originally by Melzack and Wall, the gate control theory posits presynaptic gating of C-fibre terminals by Aβ afferents, and vice versa, via an interceding GABAergic interneuron [29]. Direct recordings from neurons in superficial lamina of the dorsal horn have verified aspects of this model. For instance, electrical stimulation of large calibre Aβ, and particularly Aδ, dorsal root afferents inhibited EPSCs evoked by C-fibre input to lamina I neurons [30]. Presynaptic inhibition is also evident in a homotypic manner between C-fibres [30] and selective optical activation of NaV1.8-positive neurons produces GABA-dependent presynaptic depolarization in NaV1.8-positive neurons [11]. In the case of nerve injury, presynaptic inhibition via GABA_A_ in NaV1.8-positive primary afferents has been implicated in the development of heat hyperalgesia [5]. However, the extent to which presynaptic GABAergic effects on nociceptive NaV1.8 sensory afferents may change with inflammation and the ensuing effect on mechanical sensitivity has not been established.

Here, we used conditional deletion of functional GABA_A_ in NaV1.8-positive sensory neurons to examine the role of presynaptic inhibition in the development of hypersensitivity following inflammation with complete Freund’s adjuvant (CFA). Disruption of presynaptic inhibition blunted the development of punctate mechanical allodynia following CFA and this phenotypic deficit was paralleled by a reduction in the number of *c-fos*-positive dorsal horn cells activated by punctate but not dynamic stimuli. Dual-labelling of *c-fos* mRNA and c-Fos protein revealed that the percentage overlap of cellular circuits in the dorsal horn activated by dynamic and punctate stimuli was not changed and, thus the deficit in punctate responsiveness was due to a general reduction in synaptic input to the spinal cord after CFA. GABAergic presynaptic inhibition is thus required for the full development of punctate mechanical allodynia following CFA inflammation in glabrous skin.

## 2. Materials and Methods

### 2.1. Ethics Approval

In accordance with EU and national regulations for animal experimentation, approval from the local ethics committee for behavioural experiments was provided under approval number 35-9185.81/G-128/17 (Regierungspräsidium Karlsruhe, Germany). Approval for post-mortem collection of segments of sural nerve was provided by the Ethics Committee of the Medical Faculty Mannheim under approval numbers I-19/05 and I-21/05.

### 2.2. Animals

Wildtype (WT) C57BL/6N mice of both sexes were purchased from Charles River Laboratories (Écully, France). Mice with conditional deletion of the *gabrb3* gene encoding the β3 subunit of GABA_A_R were generated by crossing *Gabrb3^fl/fl^* (purchased from JacksonLabs#008310) with *Scn10a^Cre^* (generously provided by Dr. Rohini Kuner) [31]. The resulting *Scn10a^Cre^*;*Gabrb3^fl/fl^* line has previously been validated [5]. Progeny from this line were viable and sensory or motor phenotypes are detailed below. Control mice comprised wildtype and littermate *Gabrb3^fl/fl^*.

### 2.3. Genotyping

Genomic DNA was isolated from ear punch biopsies taken for identification and genotyping was performed by standard PCR. The primer sequences were: forward 5′-3′ ACA GAC ATA CTC ATA ATA TTT CTG TGA TT and reverse 3′-5′ GCT GAG TGC AGA CAT TCT TAC CC for *Scn10a^Cre^* and forward 5′-3′ ATT CGC CTG AGA CCC GAC T and reverse 3′-5′ GTT CAT CCC CAC GCA GAC for *Gabrb3^fl/fl^.*

### 2.4. Anaesthesia and Housing

Mice were anaesthetised with volatile anaesthetic (sevoflurane or isoflurane) before being killed by cervical dislocation. Inhalation anaesthesia was achieved by placing individual animals in a glass chamber filled with room air together with gauze soaked in anaesthetic. Anaesthetic depth was monitored periodically by corneal blink and paw withdrawal reflexes.

Mice were group housed (maximum 4) in ventilated polycarbonate cages (GM500SU, Tecniplast, Hohenpeissenberg, Germany) with access to food and water ad libitum. The housing facility was maintained at 20 °C with a 12-h light dark cycle. Behavioural experiments were performed in groups of 12 mice (6 male and 6 female) with ages ranging from 10–20 weeks. For excitability testing, 12 mice, all male, were used with an average age of 5.7 ± 3.4 months and an average of body weight of 26.7 ± 4.9 g.

### 2.5. Characterization of Sensory and Motor Function in Transgenic Mice

Homozygotic *Gabrb*3 deletion leads to epileptic seizures and demonstrable motor deficits with myoclonus [32]. Similarly, heterozygotic *Gabrb*3 deletion increased seizure susceptibility. Lower thresholds to noxious heat, as well as lower withdrawal thresholds to innocuous punctate mechanical stimuli were also evident, albeit only in male mice, particularly those lacking the paternal allele [33]. We therefore tested motor and basal sensory function in conditional *Scn10a^Cre^*;*Gabrb3^fl/fl^* mice. Basic motor performance was assessed using voluntary physical activity, locomotor activity and dynamic weight-bearing tests that were each performed over three consecutive days. Behavioral sensory testing was tested over four days with fur clip and sticky tape tests of low-threshold touch performed on day 1, von Frey and Hargreaves tests on day two, hot plate and cold plate tests on day 3, and acetone and pinprick tests on day 4. For all behavioral tests, the experimenter performing the test was blind to genotype.

### 2.6. Assessment of Sensitivity to Light Touch

Fur clip and sticky tape tests were used to assess sensitivity to light touch. Mice were acclimatised to a 20 × 20 cm transparent acrylic chamber for 20 min after which a 3 mm wide alligator clip was fastened to a clump of hair on the animal’s back just above the tail. The time (<180 s) between clip application and an overt attempt to try to remove it was recorded. The clip was applied three times at 5 min-intervals and the average response latency determined for each animal. A second test of light touch comprised a modified version of the original sticky tape test [34] for which a 1 × 3 cm strip of adhesive tape (3M, MM1530) was gently affixed along the dorsal aspect of the tail. The latency in seconds (<300 s) until the first overt attempt to try to remove the tape was recorded only as a single trial.

### 2.7. Hot/Cold Plate Assessment of Sensitivity to Noxious Hot and Cold

Sensitivity to noxious hot and cold was determined using fixed temperature hot and cold plates (Dynamic Cold/Hot Plate, Bioseb, Vitrolles, France). Mice were placed into a clear acrylic chamber with the floor plate regulated at either 54 °C (hot) or −5 °C (cold). The latency to flinching, jumping, or paw licking was recorded and an upper cut-off time of 30 s was set to avoid tissue damage. Each animal was tested once.

### 2.8. Assessment of Sensorimotor Function

Sensorimotor testing was performed at the Interdisciplinary Neurobehavioral Core (INBC) within the Medical Faculty of Heidelberg University and four test paradigms were used.

For the rotarod test of motor performance, mice were placed upon a 5.7 cm diameter, 25 cm long rod that rotated about its long axis at a speed of 20 rpm (Ugo Basile, Gemonio, Italy) for 5 min. The time between placement on the rod and the animal falling off was recorded.

Voluntary wheel running (Lafayette Instrument, Lafayette, LA, USA) was used to quantify voluntary physical activity. Mice were placed in a cage equipped with a running wheel. The time spent running and the speed of the wheel during running were recorded over 24 h and analysed using AWM software (Activity Wheel Monitor, Version 10.4. Lafayette Instrument, IN, USA).

To quantify exploratory and locomotor activity, an automated Laboratory Animal Behavior Observation Registration and Analysis System (Laboras, Metris, Hoofddorp, The Netherlands) was used to record total distance, average and maximal speeds, rearing, locomotion, climbing, grooming, and periods of immobility over 24 h. Testing began at 8 a.m. and each animal was tested individually.

Gait and locomotion were quantified using the CatWalk XT version 10.6 system (Noldus, Wageningen, The Netherlands). The mouse was placed at the entrance to a tunnel, through which it could voluntarily traverse a 1.3 m enclosed glass-bottomed corridor that allowed foot strikes to be monitored optically. Individual mice performed three trails and maximum variation in walking speed, average walking speed, duration of run, stride length, duration of standing, and swing phase (i.e., the duration of absence of contact of the paw with the glass plate) were recorded and analysed using CatWalk XT version 10.6 gait analysis software.

### 2.9. Induction of CFA Inflammation

Under 3% isoflurane anaesthesia, complete Freud’s adjuvant (CFA, F5881, Sigma-Aldrich GmbH, Taufkirchen, Germany) was injected intradermally into the metatarsal skin of the left hind paw as a 20 µL bolus using a 25-gauge needle and Hamilton syringe. After insertion, CFA was injected and the plunger held in place for 10 s to maintain pressure before withdrawing [35]. For the sham control, 20 µL saline was used instead of CFA. Animals were returned to their cage to recover for at least 24 h before being subject to any behavioral tests.

### 2.10. Assessment of Mechanical and Thermal Sensitivity after CFA

Prior to behavioral testing, animals were habituated to the testing environment by being placed in the test cage for at least 30 min on each of four consecutive days before baseline measurements. Behavioral indices of mechanical and thermal sensitivity were determined on day 0 (baseline, before injection) and then daily, for days 1–7, and subsequently on days 14, 21, and 28. Cages with a wire mesh floor were used to access the glabrous skin on the plantar surface of each hindpaw.

Sensitivity to punctate mechanical stimuli was determined with von Frey filaments ranging in strength from 0.008 to 1.4 g scaled in bending force logarithmically (Bio-VF-M, Bioseb, Vitrolles, France) and applied in ascending order, five times to each hindpaw, with an interval of at least 20 s between prods. Lifting the paw during or up to 5 s after the prod was scored as a positive response. Threshold was taken as the 60% value by interpolation of a plot of percentage of responses against filament strength.

Dynamic mechanical sensitivity was assessed with either a frayed cotton swab or a small paint brush (No. 0, da Vinci, Nürnberg, Germany), each applied in a single continuous motion beginning at the heel and moving toward the toe. Behavioral responses were given a binary score according to whether or not the animal lifted, shook, or licked the paw. Stimuli were applied 10 times with an interval of 5 min between applications. A scoring system was also used to quantify the intensity of a response and served as an index of dynamic mechanical allodynia. Scores reflected: 0, no response or lifting of the paw briefly; 1, repeated leg lifting or a single flinch; 2, leg lifted and moved laterally or tucked up into the body or a jump; 3, repeated flinching or licking of the paw.

To establish noxious mechanical threshold, an insect pin (tip diameter approximately 0.03 mm, Austerlitz insect pins; Slavkov u Brna, Czech Republic) glued to the tip of a 1 g von Frey filament was applied carefully to the plantar surface of the left hind paw 10 times at 5 min intervals. This did not injure or pierce the skin. A binary scoring system assessed nocifensive or aversive behavior and comprised paw lifting, shaking, or licking during or within 2 s of stimulus application.

A drop of acetone served as an innocuous cooling stimulus. A droplet was allowed to form at the Luer nozzle of a 2 mL syringe and was dabbed gently onto the plantar surface of the hindpaw. Behavioural responses occurring within 1 min of the cold stimulus were scored as the maximum value according to the following schema: 0, no response; 1, brief paw lifting, sniffing, flicking, or startle; 2, jumping, paw shaking; 3, multiple paw lifts, paw licking; 4, prolonged paw lifting, licking, shaking, or jumping; and 5, guarding of the paw.

Noxious heat threshold was assessed with a Hargreaves apparatus (Ugo basile, Germonio, Italy) on the plantar surface of the hindpaw. Latency to withdrawal, capped at a maximum exposure time of 20 s to prevent tissue injury, was averaged across three trials each separated by 5 min.

### 2.11. Mechanical Stimulus Protocol following CFA Inflammation

The stimulus paradigm for induction of the immediate early gene mRNA (*c-fos*) and protein (c-Fos) comprised the application of dynamic and punctate mechanical stimuli at a defined time interval. A bout of dynamic mechanical stimulation entailed 300 swipes with a cotton swab. Similarly, punctate mechanical stimulation was delivered as 300 prods with a 0.07 g von Frey filament onto the plantar surface of the paw at a frequency of 0.5 Hz. The sequence and interval between these two forms of stimuli varied according to individual protocols and spinal cord tissue was collected 30 min after the second stimulus.

Nociceptive pin prick was used for validation of dual-epoch induction, that is sequential pinprick stimulation for induction of *c-fos* and c-Fos. Each stimulus bout comprised 20 prods with a pinprick von Frey filament (1 g) to the mid-plantar region of the hind paw at 0.01 Hz. This was delivered either in one bout to determine c-fos and c-Fos time profiles or in two bouts. For single bout stimulation, tissue was collected after either 15, 30, 75, 150, or 240 min. For two bouts of stimulation the interval between stimuli was 120 min and spinal cord tissue was collected 30 min after the second stimulus.

### 2.12. Preparation of Mouse Spinal Cord

At a designated time point after the last mechanical stimulus, mice were anaesthetized cardiac perfused with chilled PBS, followed by 4% PFA. Lumbar segments L3–5 were placed in 4% PFA for 2 h and transferred subsequently to 30% sucrose-PBS at 4 °C for 18 h. Coronal cryotome (CM1950, Leica Biosystmes GmbH, Nußloch, Germany) sections of 20 µm were placed onto adhesion slides (SuperFrost Plus, Thermo Fisher, Dreieich, Germany) and stored at −80 °C. All equipment was precleaned with RNaseZAP (RNaseZAP, Sigma) and reagents were made up in diethyl pyrocarbonate (DEPC)-treated PBS.

### 2.13. Dual-Labeling of c-Fos Protein and c-fos mRNA

Slices of mouse spinal cord were double-labelled with fluorescent tags using a modified version of the previously described TAI-FISH method [36]. For tyramide-amplified in situ hybridization, slides were dried for 5 min at room temperature, washed thrice each for 3 min with ice-cold PBS, and placed in acetylation buffer (50 mL water, 670 μL Triethanolamine, 72 µL 37% HCl, 125 µL acetic anhydride) for 10 min at room temperature. After a single rinse with cold PBS, cells were permeabilized with 0.3% TX100-PBS for 20 min at 4 °C. Prior to hybridization, slides were incubated in hybridization buffer (approx. 500 µL per slide) for 1 h. Hybridization probe (1:200) was added and the tissue coverslipped to hybridize overnight at 65 °C. Sense probe was applied to control slides. Post-hybridization, slides were washed twice in 2 × SSC/0.1% *N*-Lauroylsarcosine/50% formamide at 60 °C, rinsed with RNAse buffer (10 mM Tris, pH 8.0, 500 mM NaCl, 1 mM EDTA) and then digested with 20 μg/mL RNaseA in RNase buffer for 15 min at 37 °C. This was followed by washing with 2 × SSC/0.1% *N*-Lauroylsarcosine and 0.2 × SSC/0.1% *N*-Lauroylsarcosine twice for 20 min at 37 °C and then a rinse once again with MABT (maleic acid buffer with 1% of Tween 20). Following this, tissue was blocked in MABT with 10% heat-inactivated goat serum and 1% blocking reagent for 1 h at room temperature and then incubated in the same solution with a sheep antibody to digoxygenin (anti-DIG-POD, 1:1000, Roche 11207733910) at 4 °C overnight in a humidified chamber. For signal amplification, slides were washed in MABT for 30 min at least 6 times, placed in TSA buffer for 5 min at room temperature, and subsequently incubated with TSA staining solution (dilute rhodamine tyramide 1:75 in TSA buffer plus 0.001% H_2_O_2_) in the dark for 30 min at room temperature. Slides were then washed 5 times in PBST (PBS with 0.1% Tween 20) each for 10 min, at room temperature in the dark. For immunofluorescence-tyramide-amplified in situ hybridization co-staining, slices were incubated with c-Fos primary antibody (1:1000, Abcam, ab190289) in PBST at 4 °C overnight. On the fourth day, slides were washed 4 times for 5 min in 0.2% PBST and then incubated with species-specific fluorescent secondary antibodies in PBST for 1 h at room temperature. Slides were washed 3 times for 15 min with 0.2% PBST, then washed again for 10 min 3 times in PBST and finally rinsed with 10 mM Tris-HCl for 10 min before adding cover slips with Mowiol 4-88 (Merck KGaA, Darmstadt, Germany).

For immunofluorescence co-staining, the tissue was firstly washed for 10 min, 3 times with T-BST (0.05% Tween 20 and 0.05 M Tris-HCl in PBS) at room temperature in the dark, followed by incubation with c-Fos and the primary antibody in an antibody solution (T-BST with 10% horse serum) at 4 °C overnight. On the second day, the tissue was washed 3 times for 5 min in T-BST and then incubated with species-specific fluorescent secondary antibodies in antibody solution for 1 h at room temperature. Finally, slides were washed 3 times for 15 min with 0.3% T-BST, then again with T-BST 3 times for 10 min before being rinsed with 10 mM Tris-HCl for 10 min and coverslipped with Mowiol 4-88.

The following primary antibodies were used: anti-fos (1:1000, anti-rabbit, ab190289, Abcam); anti-PKC-γ (anti Guinea pig, 1:500, GP-af350, Frontier Institute); anti-calretinin (1:1000, anti-mouse, 6B3, Swant); anti-isolectin b4 (1:500, biotin conjugate, L21140, Sigma); and anti-parvalbumin (1:2000, anti-Guinea pig, GP72 Swant) in combination with the following secondary antibodies: donkey anti-rabbit (1:700, Alexa 488, ab150073, Abcam); donkey anti-guinea Pig (1:700, Alexa 647, ab150187, Abcam); donkey anti-mouse (1:700, Alexa 647, ab150115, Abcam); and streptavidin (1:1000, Alexa 405 conjugated, S32351 Invitrogen).

### 2.14. Imaging

Fluorescence images were acquired with a scanning confocal microscope (SP8, Leica, Wetzlar, Germany) using a 20× objective (NA = 0.75), controlled by LAS X Core version 3.7.4 software (Leica, Wetzlar, Germany) and using laser wavelengths of 405, 488, 552, and 638 nm in combination with filters sets for DAPI (ex BP360/403 em LP425), FITC (ex 470/40, em LP515), and TRITC (ex 540/45, em LP590). This resulted in an average Z-optical section of 15 µm and single z-plane images of 2048 × 2048 pixels processed with LAS X Core software. The c-Fos protein signal showed a nuclear staining pattern, whereas *c-fos* mRNA appeared mostly in the cytoplasm.

### 2.15. Cell Quantification and Image Analysis

To minimize bias, the experimenter was blind to genotype and images from control and experimental groups were assigned a random number prior to analysis that was decoded after analysis. Brightness and contrast were optimized for each image. Background subtraction was performed by subtracting the mean grey scale value determined from a single background ROI placed within an unlabeled region of spinal cord from each pixel in the same image. C-Fos protein and *c-fos* mRNA signals were analysed as separate images taken from the same slice using their respective excitation wavelengths. Elliptical ROIs were placed manually around identified cells. C-Fos protein signals were typically 6–8 µm in diameter and located at or near the nucleus. Nuclei were identified by DAPI staining (DAPI, Alexa-488). *c-fos* mRNA signals were evident in the cytoplasm as regions of 8–16 µm diameter. The signal intensities of mRNA and protein were typically at least 80 times higher than background and c-Fos protein. To establish co-labelling of c-Fos protein and *c-fos* mRNA, a merged channel was generated. Cells were deemed double-labelled if in the merged channel they were labeled yellow, i.e., expression of both c-Fos protein and *c-fos* mRNA. Overlap ratio values were then calculated for each image as the number of overlapped cells divided by the number of either c-Fos protein or *c-fos* mRNA-positive cells.

### 2.16. Processing of Dorsal Horn Images and Spatial Density Estimation

To determine the spatial distribution of fluorescent signals between mice, the coordinates of individual fluorescent signals were transformed onto a reference template. As shown in Figure 1, a dorsal horn image was acquired using fluorescent staining of CGRP for lamina I [37], IB4 for lamina IIo [38], and PKC-γ for lamina IIi [39] (Figure 1A). The ventral border of lamina III, was delineated as the most ventral extent of PKC-γ dendritic arbors while the division between lamina IV and V was taken from the Allen Institute atlas. This template of a spinal cord with lamina distributions indicated by CGRP, IB-4, and PKC-γ fluorescence was generated from an overlay of four individual slices of lumbar spinal cord taken from a single naïve wildtype mouse.

To align individual dorsal horn images with the template, a semi-automated, landmark-based image registration method was developed. This comprised three steps. Firstly, for each image, six landmarks were placed manually; one at the central canal and a further five equally spaced landmarks along the outer border of lamina I on the dividing line between white matter and gray matter (Figure 1B). Similarly, the same six landmarks were defined in the dorsal horn template. In a second step, a piecewise linear transformation between the annotated landmarks in individual images and those in the template was computed. In the third step, the coordinates of the fluorescent signals were transformed into the coordinate space of the template (transformation, Figure 1B). Note for dorsal horn images, this non-rigid registration method was superior to a global affine registration method. A version of the software used to perform this registration is provided as a tool entitled “Landmark Registration” available at Galaxy Europe: https://usegalaxy.eu, accessed on 20 June 2022.

After transformation of the fluorescent signals onto the template, a Gaussian kernel density estimation (σ = 40 µm) was used to determine the relative spatial frequency of cells from which a density map was generated (Density estimation, Figure 1B). This resulted in a density map that was independent of the fluorescence intensity of individual cells. 

### 2.17. Mouse Sural Nerve Preparation

Electrical excitability testing techniques have been previously described for both unmyelinated [21] and myelinated [40] axons in peripheral nerves. Briefly, ex vivo segments of mouse sural nerve were removed bi-laterally over a distance of approximately 1 cm from the exit of the sciatic nerve to its ramification at the Achilles tendon. The epineurium was removed and the nerve placed between two suction electrodes in a custom-made organ bath (volume ca. 1 mL). Each end of the nerve was drawn into a suction electrode and embedded in petroleum jelly to establish a mechanical and high-resistance electrical seal. The distance between suction electrodes varied in accord with the length of dissected nerve and was typically between 5 and 10 mm. The organ bath was perfused continuously at a flow rate of 10 mL·min^−1^ with physiological solution of the following composition (in mM): NaCl 118, KCl 3.2, HEPES 6, Na gluconate 20, CaCl_2_ 1.5, MgCl_2_ 1.0, and d-glucose 5.55, adjusted to pH 7.4 with NaOH. The solution was bubbled continuously with oxygen and the temperature held constant at 32 °C by an inline Peltier device regulated by feedback from a thermistor in the bath.

### 2.18. Compound Action Potential Recordings

Compound action potentials were recorded extracellularly from isolated segments of sural nerve using a pair of chlorided silver wire electrodes over a Vaseline seal with one electrode inside a glass suction electrode and the other in the bath. Signals were amplified (N104 Neurolog, Digitimer, Hertfordshire, UK), filtered (NL125 Neurolog, Digitimer, Hertfordshire, UK), digitized (NI-600, National Instruments, Austin, TX, USA), and processed online using QTRAC software (Prof. Hugh Bostock, Digitimer, Hertfordshire, UK). Electrical stimuli were delivered at constant current (A395, WPI, Sarasota, FL, USA) between a second pair of chlorided silver wire electrodes, one inside and the other outside the suction electrode.

To isolate compound action potentials (CAP) deriving from A-fibres (A-CAP) and C-fibres (C-CAP), the duration of rectangular electrical stimulation was 0.1 ms for the A-CAP and 1 ms for the C-CAP. To quantify the CAP, a digital time window discriminator was implemented in QTRAC software. The window restricted analysis of the signal to a time domain after the electrical stimulus. Typical values for the time window were between 0 and 4 ms for the A-CAP and between 5 and 20 ms for the C-CAP. The position of the window was adjusted for each recording, according to the length of nerve between the electrodes and the stimulus protocol. CAP responses to each electrical stimulus were analysed online to determine peak-to-peak amplitude and latency of the signal within the time window. Latency was taken as the time from stimulus onset to the first positive-crossing of either the A-CAP or C-CAP signal at an amplitude 50% of the maximum peak positive amplitude. The timing and amplitude of electrical stimulation was controlled by QTRAC software.

### 2.19. Assessment of Electrical Excitability in A-Fibres

For A-fibres in mouse sural nerve, multiple excitability measures were determined as established for clinical research [41] using the TROND stimulation protocol [40] in QTRAC. Excitability parameters were derived from this standardized TROND stimulation protocol. Strength–duration time constant was determined from the intercept on the *x*-axis from the linear regression of charge on stimulus duration, using duration values of 0.2 ms and 1 ms to evoke a 50% A-CAP. Changes during hyperpolarizing and depolarizing sub-threshold currents designated threshold electrotonus were determined at time points during a 100 ms sub-threshold pulse and over a 100 ms recovery period thereafter. Threshold–current slope was determined from a plot of change in excitability during varying amplitudes of 200 ms long hyperpolarizing the depolarizing current pulses. Finally, refractoriness at 2 ms and super excitability at 5 ms were calculated from the time-dependent recovery cycle of excitability following a supra-maximal compound action potential in A-fibres.

### 2.20. Assessment of Electrical Excitability in C-Fibres

C-fibre excitability was assessed using threshold tracking over time as described previously [21]. Briefly, the stimulus current intensity was adjusted by the QTRAC software with the aim of evoking a C-CAP of constant amplitude; specifically, a target amplitude 40% of the maximal (100%) C-CAP. Stimuli were delivered in rolling sequence of 3 stimulus conditions repeated continuously at 1 s intervals. The sequence was a supra-maximal stimulus to evoke a 100% C-CAP; an unconditioned stimulus to evoke a 40% C-CAP; and a conditioned stimulus to evoke a 40% C-CAP preceded by a conditioning supra-maximal electrical stimulus 30 ms beforehand. The stimulus current required to maintain a 40% C-CAP unconditioned and conditioned was used to calculate an excitability index defined as;
(1)$$Excitability\;index=\frac{\left(40\%\;cond.\;current-40\%\;current\right)}{40\%\;current}$$

During tracking, pharmacological agents were added to the bath perfusate by switching the intake source of the peristaltic pump from one cylinder with standard HEPES solution to another cylinder containing the substance diluted in standard HEPES solution.

### 2.21. Induction and Assessment of Activity Depending Slowing

Activity-dependent slowing (ADS) was determined to assess the ability of C-fibres to sustain firing over minutes. For this, the change in latency of the C-CAP response was determined during stimulation at 2.5 Hz for a period of 3 min following a basal stimulus frequency of 0.5 Hz. ADS was quantified by normalizing the latency of each C-CAP response to the average C-CAP latency over the preceding 30 s period immediately before stimulation at 2.5 Hz. Changes in axonal conduction latency reflect both relative changes in axonal membrane potential and sodium channel availability [42].

### 2.22. Chemicals

CFA (F5881), GABA, 1(*S*),9(*R*)-(−)-bicuculline methiodide, NaCl, KCl, 4-(2-hydroxyethyl)-1-piperazineethanesulfonic acid (HEPES), CaCl_2_, MgCl_2_, and D-Glucose were all purchased from Sigma-Aldrich GmbH, Taufkirchen, Germany.

### 2.23. Data Analysis and Statistics

Analysis of compound action potential recordings was performed in QTRAC and using custom written routines in Igor Pro 8 (WaveMetrics, Lake Oswego, OR, USA). GraphPad Prism (San Diego, CA, USA) was used for statistical analyses. Student’s *t*-test were performed for comparisons between groups of paired and unpaired data. One-way and two-way ANOVAs were used to compare more than two groups, followed by post-hoc tests with either Tukey, Dunnett, or Bonferroni correction. Values of *p* < 0.05 were considered significant and datasets are represented as mean ± s.e.m. Significances are indicated in the Figures with an asterisk as follows; * *p* < 0.05 and ** for *p* < 0.01. Values of *p* > 0.05 were considered not significant and indicated in Figures with “n.s”.

## 3. Results

The influence of presynaptic GABAergic signaling was examined by restricting deletion of functional GABA_A_ to the NaV1.8-expressing subset of primary afferents neurons. Previous reports indicate that GABA_A_-β3 mRNA and protein were both absent from DRG neurons in *Scn10a^Cre^;Gabrb3^fl/fl^* mice [5].

### 3.1. GABA_A_-β3 Deletion in NaV1.8^+^ Afferents Did Not Affect Voluntary Motor Behavior

Global deletion of GABA_A_-β3 is lethal [32] and heterozygotic deficiency leads to profound motor deficiencies with myoclonus and epileptic seizures [33]. Inducible deletion of GABA_A_-β3 in sensory neurons (*Adv**^Cre^*;*Gabrb3*^−/−^) lead to an increased exploratory behavior in an open field test and less aversion to the open arms of the elevated maze test [11]. We therefore characterized the effect of conditional GABA_A_-β3 knock-out in NaV1.8^+^ sensory neurons on motor function using voluntary wheel running. Over a 24-h observation period of facultative wheel running, no significant differences were found between *Scn10a**^Cre^*;*Gabrb3**^fl/fl^* mice and their littermate controls with regard to the total distance covered (Appendix A). We also tested passive locomotor behavior of *Scn10a^Cre^*;*Gabrb3**^fl/fl^* mice using the Rotarod test and the time spent on the rotarod was not different between knock-out mice and littermate controls (Appendix A). Innate motor behavior was monitored by tracking home cage activity over a 24-h period. *Scn10a**^Cre^*;*Gabrb3**^fl/fl^* mice were neither more nor less active than control mice as scored by duration of rearing, grooming, or duration of immobility (Appendix A). Analysis of gait parameters revealed that *Scn10a^Cre^*;*Gabrb3^fl/fl^* mice spent more time in a standing position and more time running albeit with a prolonged swing phase and at a lower average speed (Appendix A). This suggests that the conditional deletion of GABA_A_-β3 in NaV1.8^+^ DRG neurons had a minimal effect on locomotor function and voluntary movement but carried some impairment of gait, possibly due to modified motor reflexes.

### 3.2. GABA_A_-β3 Deletion in NaV1.8^+^ Afferents Increased Basal Sensitivity to Heat

Sensory alterations are also apparent in GABA_A_-β3 knock-out mice. Heterozygotic GABA_A_-β3 deletion results in shorter withdrawal latencies to noxious heat and lower thresholds for withdrawal to punctate von Frey stimulation [33]. Conditional GABA_A_-β3 deletion in *Adv*^Cre^;*Gabrb**^fl^*^/-^ [43] as well as inducible *Adv**^CreER^*;*Gabrb3^fl^*^/*fl*^ [11] and *Scn10a^Cre^*;*Gabrb3^fl/fl^* deletion [5] all exhibit an enhanced sensitivity to dynamic [11,43] and punctate mechanical stimuli, as well as to heat [5]. In contrast, we observed a slight increase in sensitivity to punctate stimuli in *Scn10a^Cre^*;*Gabrb3^fl/fl^* mice compared with littermate controls (Figure 2A, two-way ANOVA, F(1,128) = 14.74, *p* < 0.001) that was not different between male and female mice (two-way ANOVA, F(1,128) = 0.126, *p* = 0.09). Basal sensitivity to dynamic mechanical stimulation was not different between *Scn10a^Cre^*;*Gabrb3^fl/fl^* and littermate control mice whether tested with a cotton swab (Figure 2B, two-way ANOVA, F(1,172) = 15.2, *p* = 0.41) or a paint brush (Appendix A) and no differences between male and female mice were evident (Figure 2B, two-way ANOVA, F(1,172) = 35.39, *p* = 0.21). However, the threshold to noxious heat was lower in *Scn10a^Cre^*;*Gabrb3^fl/fl^* mice when assessed with the Hargreaves test (Figure 2C, two-way ANOVA, F(1,162) = 85.2, *p* < 0.001), and similarly, when tested using a 50 °C hot plate (Figure 2D, two-way ANOVA, F(1,65) = 13.76, *p* < 0.001). This reduction was independent of sex tested with Hargreaves (2-way ANOVA, F(1,162) = 0.715, *p* = 0.64) or the hotplate (two-way ANOVA, F(1,65) = 45.10, *p* = 0.46). We have no explanation for the difference in phenotype to basal punctate mechanical stimulation between our experiments showing a slight increase (Figure 2A) and those of our own previous reports indicating basal mechanical hypersensitivity [5]. A contributing factor may be age. We used mice between 10 and 20 weeks of age, while previous studies used mice aged 8–10 weeks [5]. the sensitivity to low-threshold mechanical stimulation assessed with the sticky tape test (Appendix A) and the fur clip test (Appendix A) did not differ between *Scn10a*^Cre^;*Gabrb3^fl^*^/*fl*^ and littermate controls. Similarly, the sensitivity to cooling assessed with topical acetone (Appendix A) and a 5 °C cold plate (Appendix A) did not differ between *Scn10a*^Cre^;*Gabrb3^fl^*^/*fl*^ and littermate controls. This indicates that the disruption of presynaptic GABA_A_R on NaV1.8-positive primary afferents influences basal thresholds to punctate mechanical stimulation and noxious heat stimulation.

### 3.3. GABA_A_-β3 Deletion in NaV1.8^+^ Afferents Blunted Punctate Mechanical Allodynia after CFA

The contribution of pre-synaptic GABA_A_ to the development of inflammatory pain was explored using CFA injection into the glabrous skin on the plantar surface of the left hindpaw. Consistent with numerous previous reports [44], the threshold for a withdrawal response to punctate mechanical stimulation fell precipitously in control mice in the days following CFA (Figure 2E). However, in *Scn10a^Cre^*;*Gabrb3^fl/fl^* mice, the development of punctate mechanical allodynia following inflammation was blunted (Figure 2E, two-way ANOVA, F(1,26) = 25.17, *p* < 0.001). In contrast, dynamic mechanical allodynia, tested with a cotton swab, developed equally in both *Scn10a*^Cre^;*Gabrb3^fl^*^/fl^ and littermate controls (Figure 2E, two-way ANOVA, F(1,26) = 4.678, *p* = 0.04). Similarly, pinprick hyperalgesia (Appendix A), heat hyperalgesia tested with the Hargreaves test (Appendix A), and cold allodynia assessed using topical acetone (Appendix A), all developed with a similar time course and magnitude in *Scn10a^Cre^*;*Gabrb3^fl/fl^* mice and their littermate controls.

### 3.4. Identification of Cells in the Dorsal Horn Activated by Time Separated Punctate and Dynamic Mechanical Stimuli

To explore possible mechanisms through which deletion of functional pre-synaptic GABA_A_ in nociceptive NaV1.8^+^ neurons might lead to a suppression of punctate mechanical but not dynamic mechanical stimuli (Figure 2E,F), we first considered adaptive changes in the spinal dorsal horn (SDH). Here, we wanted to map cellular circuits responding to punctate and dynamic mechanical stimuli in the spinal dorsal horn and determine their overlap. To this end, a dual-labelling technique [36] was adapted for use in individual slices of spinal cord to map transcriptional and translational dynamics of the activity-dependent immediate early gene c-fos. TAI-FISH uses fluorescence in situ hybridization to label the immediate early gene *c-fos* mRNA and tyramine-amplification fluorescence immunohistochemistry to label its cognate protein c-Fos in the same individual tissue section (Figure 3A).

We first determined the dynamics of *c-fos* mRNA and c-Fos protein fluorescent signals in the spinal dorsal horn after a single bout of noxious pinprick stimulation of the hindpaw (Figure 3A). Noxious mechanical stimulation increased the *c-fos* mRNA signal to a peak between 30 and 60 min (red, Figure 3B), while the increase in c-Fos protein had a slower time course and was maximal approximately 120 min after the bout of pin prick stimuli (green, Figure 3B). Importantly, 30 min after pin prick stimulation, the *c-fos* mRNA signal, but not c-Fos protein was evident, while 150 min after stimulation, the c-Fos signal was still prominent, but not *c-fos* mRNA (Figure 3C,D).

To validate that TAI-FISH allowed us to distinguish spinal dorsal horn responses to sequentially applied stimuli, we stimulated the mouse hindpaw with two bouts of pinprick stimulation separated by 120 min (upper schematic, Figure 3C). The comparison of the number of cells activated by each stimulus individually with those cells activated by both stimuli (Figure 3C) indicated that 30 min after pinprick stimulation only, the *c-fos* mRNA signal was evident, while 150 min after stimulation, the c-Fos protein signal predominated (Figure 3C,D). Importantly, the locations of the cells activated by two bouts of pinpricks and thus marked by *c-fos* mRNA and c-Fos protein, largely overlapped (Figure 3D). These results indicate that dual-epoch *c-fos*/c-Fos mapping offers an effective means to distinguish between cell populations in the spinal dorsal horn responding to either or both of two time-separated peripheral stimuli.

### 3.5. Fewer c-fos-Labelled Cells in Spinal Dorsal Horn Responding to Punctate Mechanical Stimuli after CFA Injury in Mice Lacking GABA_A_-β3 in NaV1.8^+^ Afferents

Having established the utility of stimulus-dependent cell labelling in the spinal cord, the effect of CFA injection on the cellular circuitry responding to punctate and dynamic mechanical stimuli was examined (Figure 4). We first stimulated the hindpaw with a cotton swab as a dynamic mechanical stimulus and then waited 120 min before stimulating the same site with a von Frey hair as a punctate mechanical stimulus. Tissue was extracted 30 min after the second, i.e., von Frey, mechanical stimulus. According to our validation experiments (Figure 3), we regard *c-fos* mRNA signals to reflect a specific response to the punctate mechanical stimulus while c-Fos protein signals represent cells activated by dynamic mechanical stimulation. Prior to CFA injection, very few cells responded to punctate (von Frey *c-fos*, Figure 4A) or dynamic (cotton swab, c-Fos, Figure 4A) mechanical stimulation. Of those few cells that were activated under control conditions, the relative fraction of cells responding to both stimuli (Figure 4C–E) was not different between *Scn10a^Cre^*;*Gabrb3^fl/fl^* mice and littermate controls (Figure 4F,G). However, 2 days after CFA injection, robust labelling with *c-fos* mRNA and c-Fos protein was observed (Figure 4B). Quantitatively, after CFA, punctate stimulation activated an average of only 4.71 ± 1.75 *c-fos*^+^ cells per slice per mouse in *Scn10a^Cre^*;*Gabrb3^fl/fl^* mice, but this represented only a third of the 13.6 ± 5.5 average number of *c-fos*^+^ cells in the control mice (*t*-test, t(10) = 3.80, *p* = 0.0035, Figure 4H). However, cell counts for dynamic stimuli were not different between knockout and control mice with, respectively, 18.3 ± 6.2 and 15.77 ± 3.44 c-Fos^+^ cells (*t*-test, t(10) = 0.86, *p* = 0.41, Figure 4I). The counts of double-labelled cells were also significantly lower in *Scn10a^Cre^*;*Gabrb3^fl/fl^* mice at 2.44 ± 1.01 *c-fos*^+^ cells compared with 7.1 ± 2.4 cells in control mice (*t*-test, t(10) = 4,29, *p* = 0.0016, Figure 4J). The overlap of *c-fos*^+^ cells responding to punctate stimuli amongst those responding to dynamic stimuli (c-Fos^+^ cells) was similar in control and knockout mice (Figure 4K), while the fraction of punctate-sensitive cells also responding to dynamic stimuli (c-Fos) was demonstrably lower in *Scn10a^Cre^*;*Gabrb3^fl/fl^* mice (74.8 ± 16.6% cf. 30.22 ± 11.22 (t(10) = 5.45, *p* < 0.001, Figure 4L)). In total, 396 cells were demarcated as double-labeled amongst 1380 activated cells (upper, Figure 4M) indicating that after CFA, the dorsal horn pathways coding punctate and dynamic mechanical stimuli are intermingled, with approximately 28% of cells common to both pathways. This indicated that 2 days after CFA inflammation, the absence of functional pre-synaptic GABA_A_R in NaV1.8^+^ nociceptors rendered von Frey punctate stimulation less effective in activating dorsal horn cellular circuits.

Low-threshold mechanoreceptors in the skin that respond to light dynamic and punctate mechanical stimuli terminate in the LTMR-recipient zone spanning laminae III–V [45] while projection neurons reside predominantly in lamina I [46]. To explore whether the reduction in cellular activation to punctate mechanical stimuli after CFA in *Scn10a^Cre^*;*Gabrb3^fl/fl^* mice was differentially affected between dorsal horn laminae, a breakdown of the laminar distribution of *c-fos* mRNA- and c-Fos protein-positive cells by lamina was generated (Figure 5B). This showed that the *c-fos* mRNA signal to punctate mechanical stimulation in *Scn10a^Cre^*;*Gabrb3^fl/fl^* mice after CFA was reduced significantly across all laminae (upper panel, Figure 5A, two-way ANOVA, F(1,10) = 14.47, *p* = 0.0035). Importantly, only lamina III (post-hoc Bonferroni, *p* = 0.0245) and not lamina I (post-hoc Bonferroni, *p* = 0.127) showed a significant reduction in cell count.

While no differences were evident for cell counts in response to dynamic stimulation, the fraction of cells activated by both dynamic and punctate stimuli as indicated by overlapping *c-fos* mRNA and c-Fos protein signals was also significantly reduced across all laminae in *Scn10a^Cre^*;*Gabrb3^fl/fl^* mice after CFA (lower panel, Figure 5C, two-way ANOVA, F(1,10) = 18.41, *p* = 0.0016). To visualize the location of *c-fos* mRNA- and c-Fos protein-positive cells within the dorsal horn, cellular positions within each individual tissue slice were projected onto a standardized mouse spinal cord section using a mathematical transformation based on landmarks at the central canal and along the outer dorsolateral lamina I border (see methods and Figure 1B). The resulting spatial distribution of cellular circuits activated by punctate (upper, Figure 5C) and dynamic (centre, Figure 5C) mechanical stimulation are clustered in the medial aspect of the dorsal horn consistent with projection from the foot [47]. The prominent reduction in cells activated in response to punctate mechanical stimulation 2 days after CFA in *Scn10a^Cre^*;*Gabrb3^fl/fl^* mice (Figure 4) was observed in all spinal dorsal horn laminae (Figure 5) and this supports the idea that input to the spinal cord is reduced in *Scn10a^Cre^*;*Gabrb3^fl/fl^* mice lacking axonal GABA_A_. One interpretation of this could be a reduction in synaptic efficacy of NaV1.8-positive primary afferent presynaptic terminals. Another possibility is a reduced capacity of LTMRs to encode or sustain action potential firing in response to punctate mechanical stimulation.

### 3.6. Fewer Calretinin-Positive and Fewer Parvalbumin-Positive Cells Responded to Punctate but Not Dynamic Mechanical Stimuli after CFA Injury in Mice Lacking NaV1.8^+^ Pre-Synaptic GABA_A_

Interneurons in the spinal dorsal horn expressing calretinin (CR, previously known as calbindin 2) receive input from C-fibre primary afferents and chemogenetic activation of CR neurons produces punctate mechanical allodynia in the absence of injury [48]. We therefore examined whether, after CFA, the activation of CR-positive neurons was differentially affected by mechanical stimulation in the absence of GABA_A_-β3. Consistent with the general reduction in cellular activation to punctate stimulation, the number of *c-fos* mRNA^+^CR^+^ cells responding to punctate mechanical stimulation was significantly reduced in *Scn10a^Cre^*;*Gabrb3^fl/fl^* mice (Figure 6A,C, *t*-test, t(10) = 4.99, *p* < 0.001) while no differences in the number of c-Fos^+^CR^+^ cells was observed for dynamic mechanical stimulation in *Scn10a^Cre^*;*Gabrb3^fl/fl^* mice, compared to littermate controls (Figure 6A,D, *t*-test, t(10) = 0.97, *p* = 0.35). The percentage of *c-fos* mRNA-positive cells within the CR population was reduced significantly in *Scn10a^Cre^*;*Gabrb3^fl/fl^* mice (overlap in CR, Figure 6E, *t*-test, t(10) = 5.22, *p* < 0.001), while the fraction of c-Fos^+^ protein cells within the CR population was not altered in the absence of presynaptic GABA_A_ (Figure 6F, *t*-test, t(10) = 1.93, *p* = 0.083).

GABAergic inhibitory interneurons in lamina IIi and III that express the calcium-binding protein parvalbumin (PV) form axo-axonic synapses onto primary afferent terminals of low-threshold A-fibre mechanoreceptors from glabrous skin [16] and are thus prime candidates as mediators of presynaptic inhibition. Fewer cells were co-stained for PV and c-Fos protein in response to punctate von Frey stimulation after CFA injury (Figure 6B) in *Scn10a^Cre^*;*Gabrb3^fl/fl^* mice, compared to littermate controls (Figure 6G, *t*-test, t(6) = 2.40, *p* = 0.053), but the difference was not significant. However, the percentage of c-Fos protein-positive cells within the PV population was reduced significantly in *Scn10a^Cre^*;*Gabrb3^fl/fl^* mice (overlap in PV^+^, Figure 6H, *t*-test, t(6) = 2.57, *p* = 0.042) while the fraction of PV^+^ cells within those responding to punctate mechanical stimulation was not altered in the absence of presynaptic GABA_A_ (overlap in c-Fos^+^, Figure 6I, *t*-test, t(6) = 0.57, *p* = 0.586). These results suggest that after CFA, von Frey stimulation generally activated fewer dorsal horn cells in mice lacking GABA_A_-β3, but neither CR^+^ nor PV^+^ neurons were differentially affected.

### 3.7. Loss of Axonal GABA_A_ in NaV1.8^+^-Afferents Did Not Affect A-Fibre Excitability but Reduced the Capacity of C-Fibres to Sustain Action Potential Firing

One possibility to account for the generalized reduction in the number of dorsal horn cells activated by punctate mechanical stimulation is that transmission of action potential traffic from the hindpaw to the spinal cord is compromised in mice lacking axonal GABA_A_. To examine this possibility, the electrical excitability of A-fibre and C-fibre was assessed in isolated segments of mouse sural nerve from littermate control and *Scn10a^Cre^*;*Gabrb3^fl/fl^* mice.

We first assessed the excitability of A-fibres in vitro using the validated clinical technique of threshold tracking [49] to quantify indices for the action potential threshold, sub-threshold accommodation, and the time-course of threshold recovery following an action potential (Appendix A). The strength–duration time constant as an index of threshold (Appendix A), the slope of the current–threshold relationship (Appendix A), the accommodation to sub-threshold depolarizing (Appendix A) and hyperpolarizing current (Appendix A), as well as axonal refractoriness at 2 ms (Appendix A) and super-excitability at 5 ms (Appendix A), were all not different between *Scn10a^Cre^*;*Gabrb3^fl/fl^* mice and littermate controls. Similarly, no apparent differences in A-fibre excitability parameters were observed 2 days after CFA inflammation (Appendix A–S). Thus, although NaV1.8 is expressed in some A-fibres [50], these results indicate collectively that the A-fibre axonal excitability in control mice was not altered by CFA, nor did A-fibre excitability parameters differ in *Scn10a^Cre^*;*Gabrb3^fl/fl^* mice.

To assess the effect of CFA and GABA_A_R deletion on C-fibres, we assessed electrical excitability in an ex vivo sural nerve preparation. Previous in situ recordings indicate that the axonal conduction velocity of A- and C-fibres increased 4 days after CFA inflammation [51], while in isolated tissue preparations at room temperature no change in the electrical threshold or average conduction velocity was evident after CFA inflammation [4,52]. Similarly, in our ex vivo sural nerve preparation, we did not detect any changes in the amplitude of the C-fibre compound action potential (Figure 7A,B) or the basal axonal conduction velocity (Figure 7C) in response to CFA inflammation.

In DRG neuronal cell bodies, depolarizing GABA currents increase in amplitude after acute exposure to inflammatory mediators [53], likely due to an increase in intracellular chloride [27]. For persistent inflammation, 3 days after CFA injection, GABA-evoked calcium responses in DRG neurons are also increased [28]. However, whether similar changes occur in unmyelinated axons has not been determined. We previously demonstrated that GABA depolarized C-fibres via GABA_A_ and that axonal GABA responses increased within minutes of acute exposure to inflammatory soup [21]. We confirm here that GABA increased excitability in sural nerve C-fibres (100 µM and 1 µM, Figure 7D) and extend this to show that two days of CFA inflammation, axonal GABA responses are also enhanced (ANOVA, F(1,8) = 39.26, *p* < 0.01, post-hoc contra vs. CFA, t = 5.94, *p* < 0.01, Figure 7D,E). The increase in axonal GABA responses during both acute and chronic inflammation is consistent with previous reports in DRG neurons [28,54] and suggests that axonal chloride increased in the days after CFA inflammation.

To assess the potential functional impact of CFA inflammation on unmyelinated peripheral axons, we monitored the activity-dependent slowing (ADS) of conduction velocity (Figure 7F–I). ADS is an index of the ability of sural nerve C-fibres to maintain excitability and sustain firing at 2.5 Hz for 3 min. In control mice, with intact axonal GABA_A_-R, ADS was reduced by GABA (1 mM, green, contra, Figure 7F, two-way ANOVA F(1,16) = 12.21, *p* < 0.01; post-hoc t(5)= 3.25, *p* = 0.03, Figure 7G). Similarly, on the CFA-injected side, ADS was also reduced compared to the contralateral side (Figure 7G, t(16)= 3.25, *p* = 0.03); however, GABA applied to the CFA nerve did not further reduce ADS (CFA, Figure 7G, two-way ANOVA, interaction F(1,16) = 0.41, *p* = 0.53; post-hoc t(16)= 2.19, *p* = 0.26). This suggests that inflammation reduces ADS by a process that is not additive with GABA. In a separate cohort of control mice, CFA also reduced ADS (black and grey bars, Figure 7H,I, t(16) = 3.15, *p* = 0.04, Figure 7I), but in *Scn10a^Cre^*;*Gabrb3^fl/fl^* mice, CFA was without effect on ADS (red and orange bars, Figure 7I, t(16) = 0.77, *p* > 0.99). This suggests that the reduction in C-fibre ADS associated with chronic inflammation is mediated by GABA_A_-R and that protracted firing in NaV1.8^+^ C-fibre axons is compromised in its absence.

## 4. Discussion

Targeted deletion of functional GABA_A_ from NaV1.8^+^ primary sensory neurons showed that presynaptic GABAergic inhibition in NaV1.8^+^ neurons is required for punctate mechanical allodynia following inflammation. In the absence of presynaptic GABA_A_, there was reduction in the number of dorsal horn cells labelled with the activity marker *c-fos* in response to punctate von Frey but not dynamic brush stimulation following CFA. The reduction in labelled cells within the spinal circuit for punctate mechanical stimuli was evident across all dorsal horn laminae I–V and equally in cells co-stained for parvalbumin and calretinin. GABA_A_ deletion in NaV1.8^+^ neurons thus resulted in a generalised reduction in afferent activity entering spinal cord circuits following CFA. We ascribe this effect to a reduction in the efficacy of presynaptic release from NaV1.8^+^ afferents in the dorsal horn during punctate stimulation. In response to inflammation, the intracellular chloride concentration in DRG neurons increases and similarly we show that excitability responses to GABA in C-fibre axons also increase. We posit that larger axonal depolarization to GABA after CFA allows LTMR-driven presynaptic GABAergic inputs to evoke transmitter release from NaV1.8^+^ afferents. In the days following inflammation, with chloride-elevated, presynaptic GABA_A_ provides an “open gate” pathway from LTMRs to projection neurons via GABAergic interneurons, allowing punctate mechanical stimulation to co-opt dorsal horn circuits otherwise only activated by NaV1.8^+^ nociceptive C-fibres.

Diminished punctate mechanical allodynia and the reduced activation of dorsal horn circuitry both suggest that a lack of GABA_A_-R in NaV1.8^+^ afferents resulted in less synaptic input to the dorsal horn in response to punctate stimulation after CFA inflammation. We considered two possibilities that might account for this. Firstly, a reduction in primary afferent excitability. This could lead to more modest barrages in LTMR afferents responding to punctate stimulation. However, we and others [4,52,55] have found no evidence for any change in the electrical activation threshold, average conduction velocity or excitability parameters of sural nerve myelinated Aβ and Aδ fibres. It is, therefore, unlikely that the afferent activity in LTMRs is dramatically altered. However, excitability measures o unmyelinated C-fibre axons are altered following CFA inflammation. The degree of activity-dependent slowing (ADS) of conduction velocity in C-fibres was demonstrably less following CFA inflammation in control mice (Figure 7I) A reduction in ADS has been reported previously in young rats 2–5 days after CFA [55]. Here, we show that this reduction in ADS following CFA does not manifest in *Scn10a^Cre^*;*Gabrb3^fl/fl^* mice and thus requires axonal GABA_A_-R. Functionally, a reduction in ADS after CFA results in less temporal dispersion in C-fibre-evoked EPSCs in NK1-R lamina I neurons [55] and an ability to transmit higher action potential frequencies in C-fibres [51]. In control mice, both processes are likely to lead to enhanced pain behavior and may contribute to the lack of development of punctate mechanical allodynia in *Scn10a^Cre^*;*Gabrb3^fl/fl^* following CFA. Mechanistically, the basis of the reduction in ADS in C-fibres after inflammation has not been determined. However, this could be related to an elevation of intracellular chloride that occurs in DRG neurons [56] upon increased phosphorylation of NKCC1 following inflammation [27]. Our previous work has shown that ADS is increased by both the conditional deletion of GABA_A_-R in NaV1.8^+^ neurons and the NKCC1 blockade while, akin to CFA, GABA_A_-R agonists reduce ADS [21]. This suggests a permissive role of axonal GABA_A_-R in allowing elevated NKCC1 activity and thus intra-axonal chloride to exert a stabilizing influence on axonal conduction. In support of this, NKCC1 null mice exhibit a shorter and blunted behavioural response to acute capsaicin injection [57] and also have longer response times to noxious heat [58]. So, while intracellular chloride is likely to increase in C-fibres as a direct consequence of inflammation, the effect of this elevated chloride on excitability requires a chloride conductance, conceivably GABA_A_-R. Thus, a reduction in transmission fidelity in C-fibres from the periphery in *Scn10a^Cre^*;*Gabrb3^fl/fl^* mice is likely to be a contributing factor to the phenotype of blunted punctate mechanical allodynia after CFA.

A reduction in the activation of spinal circuits in *Scn10a^Cre^*;*Gabrb3^fl/fl^* mice after CFA inflammation could also be mediated centrally through the modulation of transmitter release at the first-order synapse. Disruption of presynaptic inhibition in NaV1.8^+^ neurons prevents the development of heat hyperalgesia in a model of neuropathic pain following nerve injury [5] and this study shows that punctate mechanical allodynia is blunted in the CFA inflammatory pain model (Figure 2E). The restriction of this presynaptic influence to different modalities in different pain models points to distinct presynaptic connections between specific sets of primary afferents. It is generally accepted that transmitter release from primary afferent terminals is inhibited by presynaptic GABA_A_ activation [59] through shunting inhibition and primary afferent depolarization that can be recorded as a dorsal root potential [60]. An increase in E_Cl_ following injury would enhance primary afferent depolarization (PAD), switching it from an inhibitory influence to excitation in the form of a dorsal root reflex [61]. For this canonical presynaptic circuit nociceptive C-fibre inputs are gated by low threshold Aβ and Aδ afferents and this gating switches from inhibition to excitation following injury (Figure 8). In this context, presynaptic excitation can be seen as an opening of the gate providing Aβ/δ access to nociceptive afferents via GABA_A_ leading to release of transmitters and the post-synaptic activation of projection neurons. Interestingly, GABA excitation of DRG somata leads to calcium transients that do not require overt action potential generation [62,63], since CaV3.2 T-type calcium channels suffice [64]. Our results suggest that NaV1.8^+^ C-fibre terminals constitute the gate through which afferents responding to punctate mechanical stimulation can usurp a presynaptic GABA_A_ pathway to activate projection neurons in lamina I.

In people, dynamic and static mechanical allodynia are clearly differentiated. Dynamic mechanical allodynia is elicited by light stoking of the skin and disappears with A-fibre ischaemic block [65]. Static mechanical allodynia on the other hand can be elicited by several seconds of prolonged pressure on the skin and persists during A-fibre block [65]. In rodents, dynamic mechanical allodynia following both nerve injury and inflammation is mediated by large calibre myelinated mechanoreceptors [66]. The afferent classes that signal punctate von Frey stimulation up to approximately 1 g are less well described and likely comprise Aβ, Aδ, and possibly C-LTMRs, at least in hairy skin. Aβ and particularly Aδ afferents exert potent presynaptic influence on C-fibres [30], including a subset of NaV1.8^+^ sensory neurons [11]. This population could be MrgprD nociceptive neurons that express abundant NaV1.8 [67] and conditional ablation of MrgprD nociceptors demonstrably reduced punctate mechanical allodynia [67]. Consistent with this, the optical silencing of NaV1.8^+^ afferents in adult mice produced a reduction, albeit modest, in post CFA punctate mechanical allodynia without affecting heat hyperalgesia [68].

### Role of PV- and CR-Positive Neurons in Punctate Mechanical Allodynia

The dorsal horn circuitry activated by dynamic and punctate mechanical stimuli differs according to the type of mechanical afferents activated. In addition, modulations of dorsal horn circuits after injury also differ in accordance with the type of injury [69]. For example, parvalbumin (PV)-expressing neurons in lamina III, which are known to receive Aβ input [16] and exert inhibition on excitatory PKCγ neurons. Following nerve injury, the synaptic efficacy from PV^+^ to PKCγ^+^ neurons is reduced and the ablation of PV+ neurons elicited punctate mechanical allodynia [70]. Mechanical allodynia after inflammation, however, is associated with an increased activation of calretinin (CR)-expressing neurons [71], some of which are GABAergic [72]. For CR neurons in lamina II, 80% receive *Aδ* input and all receive peptidergic C-fibre input [69]. In turn, CR neurons can make direct connections with projection neurons in lamina I [73]. The chemogenetic activation of CR neurons suppressed punctate and dynamic mechanical allodynia following CFA inflammation but not after nerve injury [69]. Despite these differences in the roles of PV- and CR-neurons in neuropathic and inflammatory punctate allodynia circuitry, our results indicate that there were not differential changes in PV- or CR-neurons following inflammation and that a loss of presynaptic GABA simply reduced the overall activation of all neurons within the punctate mechanical circuit.

We illustrate here a behavioural phenotype in which punctate mechanical allodynia resulting from CFA inflammation is selectively abolished in mice lacking functional GABA_A_ in NaV1.8^+^ sensory afferents. Mechanistically, this phenotype is attributed to an inability of LTMRs activated by punctate stimulation to access the spinal circuitry for pain. A reduction in spinal input in the absence of primary afferent GABA effects, implies that presynaptic GABA_A_ either facilities excitation or becomes overtly excitatory in NaV1.8 afferents after inflammation. The disruption of presynaptic inhibition also revealed its role in modulating basal threshold for mechanical stimuli. Presynaptic GABA modulation is potent, able to suppress signals before entering the spinal circuits and targeted disruption of this process at, or shortly after injury, may improve pathological pain outcomes.

## Figures and Tables

**Figure 1 cells-11-02390-f001:**
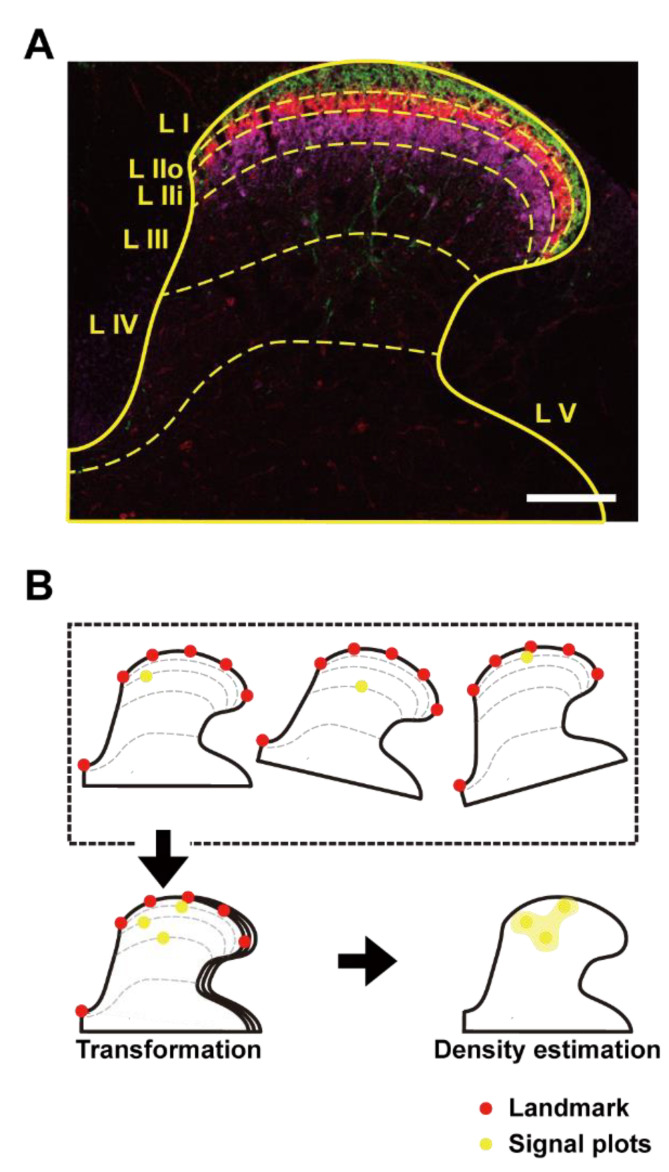
(**A**) Generation of a reference template for mouse dorsal horn with Rexed laminae delineated by immunofluorescence against CGRP (green) for lamina I, IB4 (red) for lamina IIo, and PKC-γ (purple) for lamina IIi. The scale bar indicates 100 µm. (**B**) Schematic of the transformation and spatial density estimation procedures. Initially, landmarks (red dots) were established at the central canal and equally spaced along the outer border of lamina I (upper dashed line enclosure). A piece-wise linear transformation was computed using these landmarks. This transformation was then used to map coordinates of the fluorescent signals (yellow dots) from each image into the reference spinal dorsal horn template (downward arrow). In a separate step, kernel density estimation was used generate a frequency map (rightward arrow).

**Figure 2 cells-11-02390-f002:**
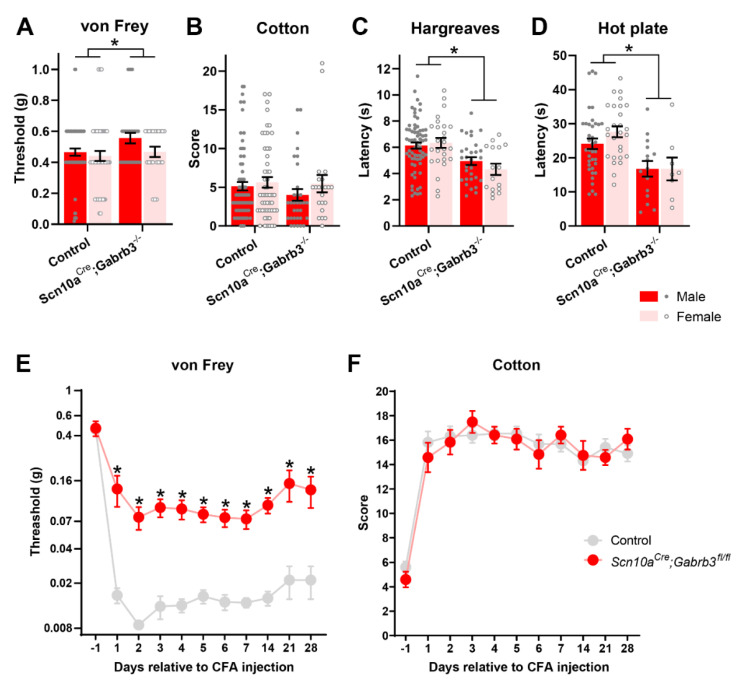
Basal sensitivity to punctate (von Frey, **A**) but not dynamic (cotton, **B**) mechanical stimuli is elevated in *Scn10a^Cre^*;*Gabrb3^fl/fl^* mice (*n* = 32 male, *n* = 22 female; right bars, (**A**) relative to littermate controls (*n* = 65 male, *n* = 52 female; left bars, **A**). Male animals are shown as closed markers and red bars (**A**–**D**) and female as open markers and pink bars (**A**–**D**). For dynamic mechanical stimuli with cotton (**B**), no differences were evident between control (*n* = 70 male, *n* = 52 female) and *Scn10a^Cre^*;*Gabrb3^fl/fl^* mice (*n* = 33 male, *n* = 22 female). Heat pain threshold determined using the Hargreaves test (**C**) was lower in *Scn10a^Cre^*;*Gabrb3^fl/fl^* mice (*n* = 30 male, *n* = 16 female) than in littermate controls (*n* = 68 male, *n* = 52 female). Similarly, for the 50 °C hot plate test (**D**), heat pain threshold was lower in *Scn10a^Cre^*;*Gabrb3^fl/fl^* mice (*n* = 15 male, *n* = 8 female) than in littermate controls (*n* = 37 male, *n* = 27 female). In the days following CFA injection into the hindpaw, a pronounced decrease in withdrawal threshold to punctate (von Frey, **E**) mechanical stimulation was observed in littermate controls (grey, **E**, *n* = 12) but not in *Scn10a^Cre^*;*Gabrb3^fl/fl^* mice (red, **E**, *n* = 22). In contrast, control (*n* = 12) and *Scn10a^Cre^*;*Gabrb3^fl/fl^* mice (*n* = 22) both developed an increased response score to dynamic (cotton, **F**) mechanical stimulation in the four-week period following CFA injection. Values of *p* < 0.05 were considered significant and are indicated with asterisks as follows; * *p* < 0.05.

**Figure 3 cells-11-02390-f003:**
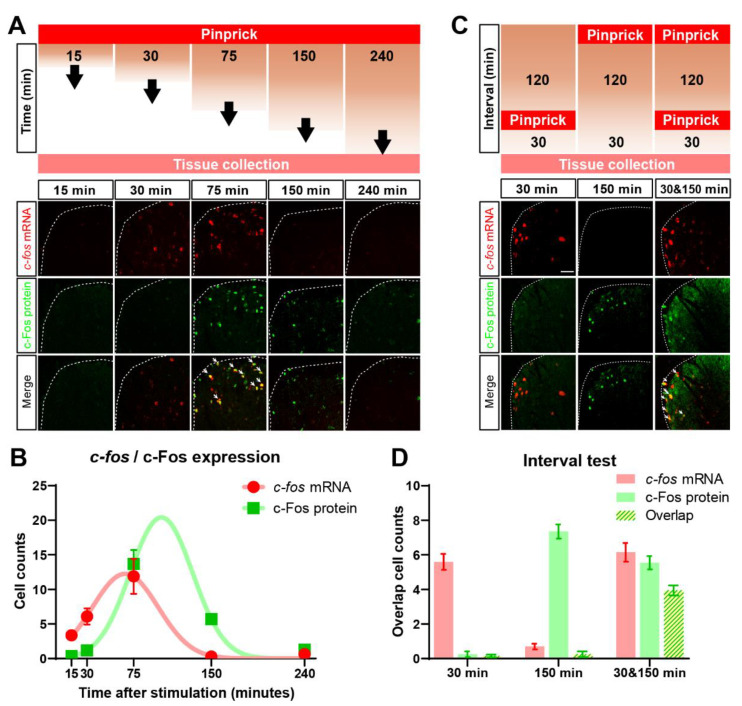
(**A**) Dual labelling of *c-fos* mRNA (red, (**B**)), c-Fos protein (green, (**A**)), and their overlap (yellow, Merge, (**A**)) in spinal dorsal horn in response to a single bout of pinprick stimulation of hindpaw glabrous skin in wild-type mice. Tissue was collected at discrete time points after stimulation (schematic, (**A**)). The time course of the number of cells positive for *c-fos* mRNA (red markers, (**B**)) and c-Fos protein (green markers, (**B**)) is estimated with Gaussian fits (red and green lines, (**B**)). Dual labelling of *c-fos* mRNA (red, (**C**)), c-Fos protein (green, (**C**)), and their overlap (yellow, Merge, (**C**)) in spinal dorsal horn in response to repeat bouts of pinprick stimulation delivered at varying inter-stimulus intervals. The number of cells activated by the first stimulus, c-Fos protein (green, (**D**)), the second stimulus, *c-fos* mRNA (red, (**D**)), and by both stimuli (yellow overlap, (**D**)) are shown at the 30 and 150 min time points. Data represent averages of 10–12 coronal slices per mouse from lumbar (L3–L5) spinal dorsal horn using 3 mice per group.

**Figure 4 cells-11-02390-f004:**
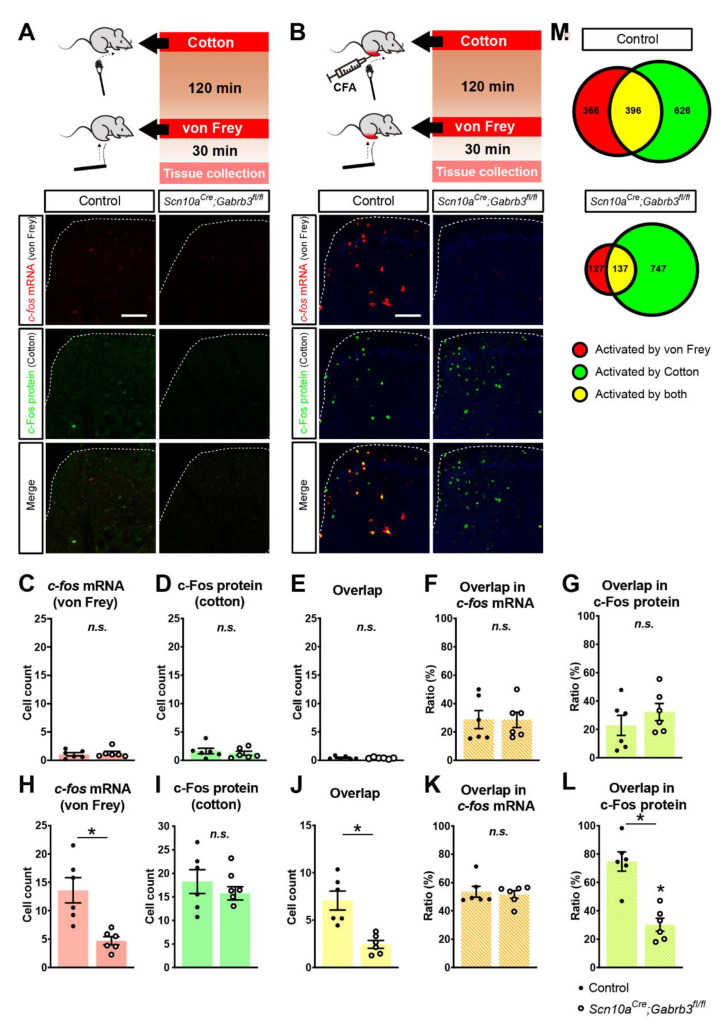
(**A**) Dual labelling of *c-fos* mRNA (red, **A**) and c-Fos protein (green, **A**) in spinal dorsal horn showed cells activated either by dynamic (cotton, **A**) or punctate mechanical stimulation (von Frey, **A**) or both (overlap, yellow, **A**) in littermate controls (left, **A**,**B**) and *Scn10a^Cre^*;*Gabrb3^fl/fl^* mice (right, **A**,**B**) before CFA. (**B**) Two days after CFA injection, the number of cells responding to dynamic (c-Fos, green, **B**) and punctate (*c-fos*, red, **B**) mechanical stimulation increased and there was also an accompanying increase in the number of cells activated by both forms of mechanical stimuli after CFA (yellow, **B**). (**C**–**L**) pooled data showing cell counts per mouse (averaged from at least 3 sections) for *Scn10a^Cre^*;*Gabrb3^fl/fl^* mice (open markers) and littermates (closed markers) prior to CFA (**C**–**G**) and 2 days after CFA (**H**–**L**). Average cell counts per mouse are shown in response to punctate (von Frey, **C**,**H**), dynamic (cotton, **D**,**I**), and both forms of mechanical stimulation (overlap, yellow, **E**,**J**). The degree of overlap is shown as the percentage of cells responding to dynamic stimulation (c-Fos) as a fraction of those responding to punctate stimulation (overlap in *c-fos*, orange, **F**,**K**) as well as the percentage fraction of cells responding to punctate stimulation (*c-fos*) amongst those responding to dynamic stimulation (overlap in c-Fos, light green, **G**,**L**). (**M**) Venn diagram depiction of the overlap of *c-fos*^+^ cells responding to punctate (red, **M**), c-Fos^+^ cells responding to dynamic mechanical stimulation (green, **M**) and their overlap (yellow, **M**) for littermate controls (upper, **M**) and *Scn10a^Cre^*;*Gabrb3^fl/fl^* mice (lower, **M**) 2 days after CFA injection. Data represent averages of 12–15 coronal slices per mouse from lumbar (L3–L5) spinal dorsal horn using 6 mice per group. Values of *p* < 0.05 were considered significant and are indicated with asterisks as follows; * *p* < 0.05, n.s. = not significant.

**Figure 5 cells-11-02390-f005:**
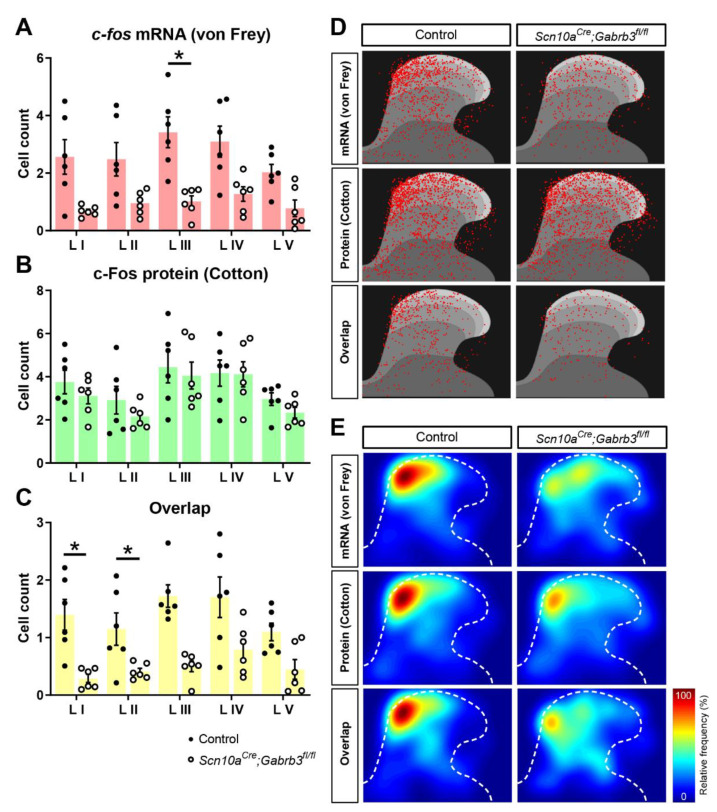
Distribution of cells positive for *c-fos* mRNA in response to punctate (red, von Frey, **A**) and c-Fos protein (green, cotton, **B**) mechanical stimulation across laminae I–V of the spinal dorsal horn for control mice (closed markers) and *Scn10a^Cre^*;*Gabrb3^fl/fl^* mice (open markers). The number of cells per mouse (cell count) labelled with both *c-fos* and c-Fos (yellow, overlap) are indicated in the lower panel (**C**). (**D**) Spatial location of cells positive for *c-fos* mRNA (upper, **D**), c-Fos protein (centre, **D**), and their overlap (lower, **D**) after transformation onto a standard spinal dorsal horn with laminae I–V shown, respectively, in progressively darker grey tones for control mice (left column, **D**) and *Scn10a^Cre^*;*Gabrb3^fl/fl^* mice (right column, **D**). (**E**) Spatial density plot of the cellular distribution in panel (**D**) for *c-fos* mRNA (upper, **E**) c-Fos protein (centre, **D**) and their overlap (lower, **D**) with spatial density increasing from blue through yellow to red for control (left column, **E**) and *Scn10a^Cre^*;*Gabrb3^fl/fl^* mice (right column, **E**). Data represent averages of 12–15 coronal slices per mouse from lumbar (L3–L5) spinal dorsal horn using 6 mice per group. Values of *p* < 0.05 were considered significant and are indicated with asterisks as follows; * *p* < 0.05.

**Figure 6 cells-11-02390-f006:**
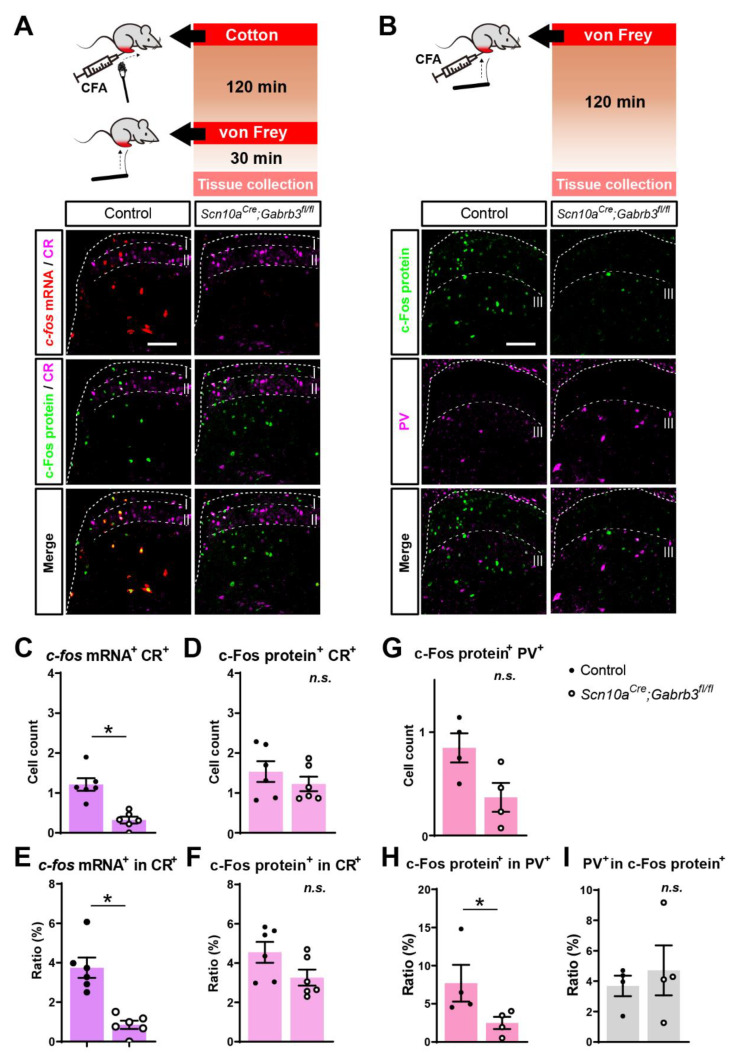
(**A**,**B**) Co-staining of calretinin-positive (CR, purple, **A**) and parvalbumin-positive (PV, purple, **B**) spinal cord neurons with *c-fos* mRNA (von Frey in schematic, **A**,**B**) and c-Fos protein (cotton in schematic, **A**,**B**)-positive cells responding to punctate (upper panel, red, **A**,**B**) and dynamic (centre panel, green, **A**,**B**) mechanical stimulation in control (left, **A**,**B**) and *Scn10a^Cre^*;*Gabrb3^fl/fl^* mice (right, **A**,**B**). The number of cells labelled with both *c-fos* and c-Fos (yellow, **A**,**B**) are indicated in the lower panel (overlap, **A**,**B**). (**C**–**F**) Counts per slice of cells positive for *c-fos* mRNA and CR^+^ (**C**) or c-Fos protein and CR^+^ (**D**) in *Scn10a^Cre^*;*Gabrb3^fl/fl^* mice (open markers, **C**–**I**) and littermate controls (closed markers, **C**–**I**). The percentage of *c-fos*-positive cells within the CR^+^ population (*c-fos* mRNA^+^ in CR^+^, **E**) and the percentage of c-Fos-positive cells within the CR population (c-Fos protein^+^ in CR^+^, **F**). (**G**–**I**) Counts per slice of cells positive for c-Fos protein and PV^+^ (**G**). The percentage of c-Fos-positive cells within the PV^+^ population (overlap in PV^+^, **H**) and the percentage of PV^+^ cells within the c-Fos population (PV^+^ in c-Fos protein^+^, **I**). Data represent averages of 12–15 coronal slices per mouse from lumbar (L3–L5) spinal dorsal horn using 6 *Scn10a^Cre^*;*Gabrb3^fl/fl^* mice and 4 littermate control mice for CR staining and 4 mice per group for PV staining. Values of *p* < 0.05 were considered significant and are indicated with asterisks as follows; * *p* < 0.05, n.s. = not significant.

**Figure 7 cells-11-02390-f007:**
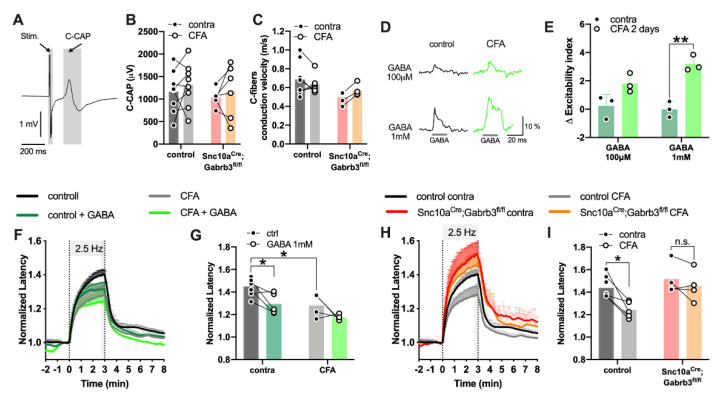
(**A**) Example sural nerve C-fibre compound action potential (C-CAP). C-CAP amplitude (**B**) and axonal conduction velocity (**C**) were not altered between sham injected contralateral (closed markers, **B**,**C**) and injected sides 2 days after CFA (open markers, **B**,**C**) in littermate controls (black and grey bars, **B**,**C**, *n* = 7) and *Scn10a^Cre^*;*Gabrb3^fl/fl^* mice (red and orange bars, **B**,**C**, *n* = 6). D, In naïve wildtype mice, axonal responses to GABA at 100 µM (upper traces, **D**) and 1 mM (lower traces, **D**) are increased 2 days after CFA inflammation (green traces, **D**). (**E**) Pooled data for the change in axonal GABA response amplitude in contralateral sural nerves (dark green bars, **E**, *n* = 3) and 2 days after CFA injection (light green bars, **E**, *n* = 3). (**F**) Electrical stimulation at 2.5 Hz for 3 min (grey shading, **F**) resulted in activity-dependent slowing (ADS) of axonal conduction velocity measured as an increase in C-CAP latency (Normalized latency, **F**). (**G**) Effect of GABA (1 mM, green, **F**) on ADS (Normalized latency, **G**) in the untreated contralateral sural nerve (contra, **G**, *n* = 6) and CFA injected side (CFA, **G**, *n* = 3). H, ADS in *Scn10a^Cre^*;*Gabrb3^fl/fl^* (red, **H**) and littermate control mice (black, **H**) before (black and red, **H**) and 2 days after CFA (grey and orange, **H**). ADS was reduced after CFA (open markers and grey bar, **I**) in control mice (*n* = 5) but not in *Scn10a^Cre^*;*Gabrb3^fl/fl^* mice (red and orange bars, **I**, *n* = 4). Values of *p* < 0.05 were considered significant and are indicated with asterisks as follows; * *p* < 0.05 and ** *p* < 0.01, n.s. = not significant.

**Figure 8 cells-11-02390-f008:**
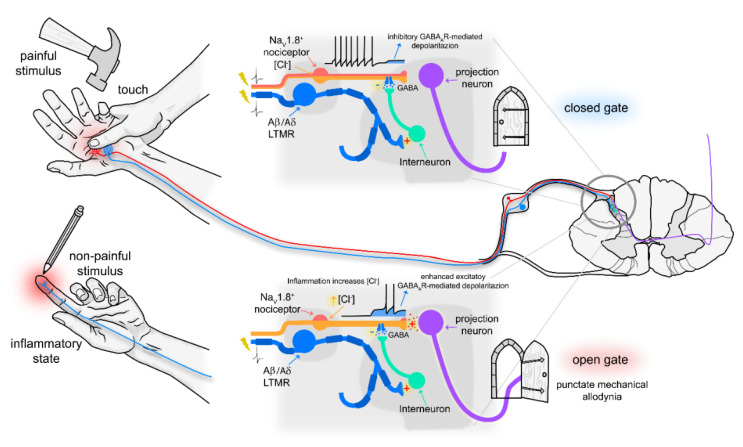
Schematic illustrating putative role of presynaptic GABA_A_-R in mediating punctate mechanical allodynia. Immediately after injury, light touch activates Aβ/Aδ afferents (blue) that can relay onto GABAergic interneurons (green) that, in turn, make presynaptic contact with NaV1.8^+^ nociceptive afferent terminals (red). In the upper panel, GABA acting via presynaptic GABA_A_-R results in modest depolarization of the nociceptive terminal (blue shading) that quells transmitter release, closing the gate to transmission from NaV1.8^+^ afferents to nociceptive projection neurons (purple). In response to persistent inflammatory injury, the intracellular chloride concentration in NaV1.8^+^ sensory neurons is elevated (orange shading). This allows GABA acting via presynaptic GABA_A_-R to elicit transmitter release from NaV1.8^+^ terminals, opening the gate for a circuit from Aβ/Aδ afferents, through GABAergic interneurons that usurp the NaV1.8^+^ to activate nociceptive projection neurons.

## Data Availability

The data presented in this study are openly available upon reasonable request.

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
