# Peer review of "Pre-Synaptic GABAA in NaV1.8+ Primary Afferents Is Required for the Development of Punctate but Not Dynamic Mechanical Allodynia following CFA Inflammation"

_cells, 2022, doi:10.3390/cells11152390_

Round 1
Reviewer 1 Report
Very interesting and well-designed study. Methods are described in adequate detail to support replication. Results were clearly presented. Conclusions were appropriate.
Minor concerns:
1. There are abbreviations used throughout the study and in many instances those are not spelled out which may be confusing to the reader. Please check that all abbreviations are spelled out the first time the abbreviation is used.
2. Please report the secondary antibodies, dilution, company, product number, etc. for the secondary antibodies used in the fluorescence microcopy study.
3. Cell Counts: how were the nuclei identified in the microscopy images (labeling with DAPI, etc.)?
4. Figures: please consider including the sample size per group in the figure descriptions.
Author Response
Minor concerns:
- There are abbreviations used throughout the study and in many instances those are not spelled out which may be confusing to the reader. Please check that all abbreviations are spelled out the first time the abbreviation is used.
Response: The abbreviations used throughout the manuscript have been checked. We hope that each abbreviation is now defined upon first use and then used in a consistent manner subsequently.
- Please report the secondary antibodies, dilution, company, product number, etc. for the secondary antibodies used in the fluorescence microcopy study.
Response: The secondary antibodies have been added to the Methods on pp. 7, line 312;
“… in combination with the following secondary antibodies: Donkey anti-Rabbit (1:700, Alexa 488, ab150073, Abcam); Donkey anti-Guinea Pig (1:700, Alexa 647, ab150187, Abcam); Donkey anti-Mouse (1:700, Alexa 647, ab150115, Abcam); Streptavidin (1:1000, Alexa 405 conjugated, S32351 Invitrogen).”
- Cell Counts: how were the nuclei identified in the microscopy images (labeling with DAPI, etc.)?
Response: We thank the reviewer for highlighting the inadequate description of the process of identification of nuclei and have added the following text;
- 7, line 335: “Nuclei were identified by co-staining with DAPI.”
- Figures: please consider including the sample size per group in the figure descriptions.
Response: We appreciate the suggestion form the reviewer to include sample sizes in the legends to each Figure and these have been added as follows;
Figure 2A, pp.13, line 535-6: “…in Scn10aCre;Gabrb3fl/fl mice (n=32 male, n=22 female; right bars, A) relative to littermate controls (n=65 male, n=52 female; left bars, A).“
Figure 2B, pp.13, line 538-9: “…cotton (B) no differences were evident between control (n=70 male, n=52 female) and Scn10aCre;Gabrb3fl/fl mice (n=33 male, n=22 female).”
Figure 2C, pp.13, line 539-41: “…Hargreaves test (C) was lower in Scn10aCre;Gabrb3fl/fl mice (n=30 male, n=16 female) than in littermate controls (n=68 male, n=52 female).”
Figure 2D, pp.13, line 542-3: “…in Scn10aCre;Gabrb3fl/fl mice (n=15 male, n=8 female) than in littermate controls (n=37 male, n=27 female).”
Figure 2E, pp.13, line 544-5: “…littermate controls (grey, E, n=12) but not in Scn10aCre;Gabrb3fl/fl mice (red, E, n=22).”
Figure 2F, pp.13, line 545-6: “In contrast, control (n=12) and Scn10aCre;Gabrb3fl/fl mice (n=22) both developed an increased response score to dynamic (cotton, F)…”
Figure 3, pp. 13, line 578: “Data represent averages of 10-12 coronal slices per mouse from lumbar (L3‑L5) spinal dorsal horn using 3 mice per group.”
Figure 4, pp.16, line 641-2: “Data represent averages of 12-15 coronal slices per mouse from lumbar (L3‑L5) spinal dorsal horn using 6 mice per group.”
Figure 5, pp. 17, line 682-3: “Data represent averages of 12-15 coronal slices per mouse from lumbar (L3‑L5) spinal dorsal horn using 6 mice per group.”
Figure 6, pp. 19, line 725-7: “Data represent averages of 12-15 coronal slices per mouse from lumbar (L3‑L5) spinal dorsal horn using 6 Scn10aCre;Gabrb3fl/fl mice and 4 littermate control mice for CR staining and 4 mice per group for PV staining.”
Figure 7b-C, pp. 21, line 760: “…grey bars, B&C, n=7) and Scn10aCre;Gabrb3fl/fl mice (red and orange bars, B&C, n=6).”
Figure 7G, pp. 21, line 763-4: “…E, n=3) and 2 days after CFA injection (light green bars, E, n=3).”
Figure 7I, pp. 21, line 770-1: “…in control mice (n=5) but not in Scn10aCre;Gabrb3fl/fl mice (red and orange bars, I, n=4).”
Reviewer 2 Report
Review: Pre-synaptic GABAA in NaV1.8+ primary afferents is required for the development of punctate but not dynamic mechanical allodynia following CFA inflammation. Cells 2022
This is a very well described study and clearly presented. The extensive details in the figures are very well delivered. The statistical details are well described, and interpretation of the data is described in a non-bias manner. It appears the literature is well covered in relation to the details of this study. This manuscript is informative and well help advance the field in pain research in neural circuitry. The overview diagram is a nice touch for the presentation. This might be a good figure for the graphical abstract.
Only minor suggestions for the authors.
Minor:
1. In Abstract: define CFA
Including a brief discussion on the significance of PV and CR positive cells (mainly CR cells as PV cells are mentioned) within the introduction may assist in clarity when reading. The authors did a wonderful job of discussing the role CR positive cells on page 17 of the results section. However, including something like this earlier in the paper may be helpful to readers.
Author Response
Minor:
- In Abstract: define CFA
Response: We have expanded CFA to “Complete Freund’s Adjuvant” in the abstract (pp. 1, line 30) and the Introduction (pp. 3, line 98).
Including a brief discussion on the significance of PV and CR positive cells (mainly CR cells as PV cells are mentioned) within the introduction may assist in clarity when reading. The authors did a wonderful job of discussing the role CR positive cells on page 17 of the results section. However, including something like this earlier in the paper may be helpful to readers.
Response: We are grateful for the helpful suggestion of the reviewer regarding details of PV and CR neuronal populations. The current manuscript provides a brief description of calretinin-positive cells in the Result on pp. 17 (line 686-688) and similarly for parvalbumin-positive cells on pp. 19 (line 700-703). There is also a complete paragraph on PV and CR subtypes in the discussion (pp.24, line900-915). However, since the findings did not show any correlation between changes in spinal cord activity patterns within either of these subpopulations, we would prefer to focus the introduction on the role of pre-synaptic inhibition in the spinal cord rather than adding discourse on cellular subtypes resident in the dorsal horn. We therefore offer a compromise by adding parvalbumin neurons to the introduction (pp. 2, line 60-63)
“comprised of inhibitory interneurons , for example, those expressing the calcium binding protein parvalbumin (PV) and parvalbumin expressing cells that form axoaxonic synapses with A-fibre mechanoreceptor afferent terminals [16].”
Reviewer 3 Report
Hypersensitivity to mechanical stimuli is a cardinal symptom of neuropathic and inflammatory pain, and Authors used conditional deletion of GABAA in NaV1.8-positive sensory neurons (Scn10aCre;Gabrb3fl/fl) to manipulate selectively presynaptic GABAergic inhibition. Data showed that the development of inflammatory punctate allodynia was mitigated in mice lacking pre-synaptic GABAA. In peripheral DRG neurons, CFA inflammation led to an increase in axonal excitability responses to GABA. In the days after inflammation, presynaptic GABAA in NaV1.8+ nociceptors constitutes an “open gate” pathway allowing mechanoreceptors responding to punctate mechanical stimulation access to DRG nociceptive circuits. Disruption of presynaptic inhibition also revealed its role in modulating basal threshold for mechanical stimuli. Presynaptic GABA modulation is po-tent, able to suppress signals before entering the spinal circuits and targeted disruption of this process at, or shortly after injury, may improve pathological pain outcomes.
The article is very complex, but the logical structure is rigorous, the experiments conducted in a linear way, the results reported in a clear and understandable way.
The bibliography is complete.
Author Response
We thank the reviewer for the support and their effort and time in reviewing the manuscript.